# Age-Specific Demographic Response of a Long-Lived Scavenger Species to Reduction of Organic Matter in a Landfill

**DOI:** 10.3390/ani13223529

**Published:** 2023-11-15

**Authors:** Diego J. Arévalo-Ayala, Joan Real, Santi Mañosa, Joan Aymerich, Carles Durà, Antonio Hernández-Matías

**Affiliations:** 1Equip de Biologia de la Conservació, Departament de Biologia Evolutiva, Ecologia i Ciències Ambientals, Facultat de Biologia, Universitat de Barcelona, Avda. Diagonal 643, 08028 Barcelona, Spain; jreal@ub.edu (J.R.); smanosa@ub.edu (S.M.); ahernandezmatias@ub.edu (A.H.-M.); 2Institut de Recerca de la Biodiversitat (IRBio), Universitat de Barcelona, 08028 Barcelona, Spain; 3Grup d’Anellament de Calldetenes-Osona (GACO), 08506 Calldetenes, Spain; jaigaco@gmail.com (J.A.); carles.dura@ornitologia.org (C.D.); 4Estació Biològica del Montseny, Institut Català d’Ornitologia (ICO), Edifici Fontmartina, 08081 Fogars de Montclús, Spain

**Keywords:** Cormack-Jolly-Seber model, Bayesian mark-recapture survival analysis, Bayesian hierarchical model, predictable anthropogenic food subsidies, individual random effect, *Gyps fulvus*, vulture conservation, permanent emigration, apparent survival

## Abstract

**Simple Summary:**

Food availability plays a significant role in modulating populations, especially in species relying on human-generated food sources like landfills. Sudden changes in food access can negatively impact vital parameters such as survival. For long-lived scavenger species, understanding how different age groups respond to these changes is essential since landfills are condemned to be closed. We studied the response in terms of apparent survival of griffon vultures to a decline in landfill organic matter after a waste treatment center became operational. The proportion of transients increased in all age groups. Survival dropped in juveniles and adult residents but increased in immature residents. These findings suggest that vultures permanently emigrated at higher rates due to intensified competition after the reduction in food. Intriguingly, immature resident vultures showed resilience, indicating the presence of high-quality individuals despite the food scarcity. The reasons behind reduced survival in adult residents during the final four study years remain unclear but are potentially linked to non-natural mortality. This research highlights the challenges facing scavengers as European landfill sites close, thereby stressing the need for food scarcity studies and timely conservation measures.

**Abstract:**

Food availability shapes demographic parameters and population dynamics. Certain species have adapted to predictable anthropogenic food resources like landfills. However, abrupt shifts in food availability can negatively impact such populations. While changes in survival are expected, the age-related effects remain poorly understood, particularly in long-lived scavenger species. We investigated the age-specific demographic response of a Griffon vulture (*Gyps fulvus*) population to a reduction in organic matter in a landfill and analyzed apparent survival and the probability of transience after initial capture using a Bayesian Cormack-Jolly-Seber model on data from 2012–2022. The proportion of transients among newly captured immatures and adults increased after the reduction in food. Juvenile apparent survival declined, increased in immature residents, and decreased in adult residents. These results suggest that there was a greater likelihood of permanent emigration due to intensified intraspecific competition following the reduction in food. Interestingly, resident immatures showed the opposite trend, suggesting the persistence of high-quality individuals despite the food scarcity. Although the reasons behind the reduced apparent survival of resident adults in the final four years of the study remain unclear, non-natural mortality potentially plays a part. In Europe landfill closure regulations are being implemented and pose a threat to avian scavenger populations, which underlines the need for research on food scarcity scenarios and proper conservation measures.

## 1. Introduction

Food availability is a key factor that shapes demographic parameters and influences the life-history evolution of vertebrates by modulating survival, breeding performance, and, ultimately, population dynamics [1]. In general, animals may exhibit improved survival and reproductive rates when food is abundant [2,3]. Conversely, when food is limited, density-dependent processes may operate and increase both intraspecific competition and emigration probability, resulting in a reduction in the apparent survival probability of individuals [1,2,4,5]. Food can also influence age-dependent survival of species. Non-adult individuals naturally exhibit lower survival performance than adults and survival rates usually improve with age due to the selective disappearance of poor-quality phenotypes and an age-related increase in competence [6,7], which is followed by a progressive decline in survival with age due to physiological senescence [8,9]. While acquiring essential foraging and competitive skills, early-age individuals tend to exploit predictable and abundant food resources as density-dependence is relatively low [10,11,12,13,14]. However, during food shortages, density-dependence intensifies and increases the likelihood of early-age individuals becoming transients (i.e., emigrating permanently from the site) [1,4,14].

Ecosystems worldwide have been modified by human food subsidies, which have had the greatest impact in regions where most food is wasted [15]. Food subsidies such as fisheries discards, supplementary feeding stations or landfills are abundant and highly predictable in space and time (also known as predictable anthropogenic food subsidies, hereafter PAFS) and attract large numbers of species [15,16,17]. Several studies have shown that PAFS can enhance the survival probability of local populations and, above all, that of young individuals [15,18,19,20]. However, a progressive or drastic food reduction can occur in PAFS due to the application of local or regional regulations that may have negative demographic consequences for species exploiting these resources. Good examples include sanitary regulations that prohibited leaving cattle carcasses in the wild or in supplementary feeding sites for scavengers during the bovine spongiform encephalopathy (BSE) outbreak [21,22,23,24], the establishment of trawling moratoriums [1], and, more recently, the reduction in organic waste and the closure of landfills due to the planning of the European circular economy [14,25,26], which are all evidence of the negative effects of food limitation on survival and population dynamics. Even so, the age-specific demographic response to a depletion of food subsidies is still poorly understood [14,23], particularly in PAFS such as landfills.

Landfills acting as PAFS represent a continuous food source that benefits numerous opportunistic species [17]. This surplus food enhances population survival by providing sustenance year-round and, importantly, for specific age cohorts during periods of natural food scarcity (e.g., juvenile vultures during winter since the use of PAFS is energetically less demanding than searching for wild carrion) [15,27,28,29,30,31]. Yet, feeding in landfills though is hazardous, and may only provide low-quality and polluted food, with the associate risk of exposure to solid waste ingestion (e.g., plastic, rubber, glass or metals) and pathogens [32,33,34,35]. In addition, in some ecological systems and species, feeding on landfills could have a detrimental effect on chick and juvenile survival since individuals of these ages are more susceptible than adults to the abovementioned risks (e.g., [36,37]. Nevertheless, for long-lived scavengers such as vultures, the overall trade-off seems to be positive, as current evidence supports the idea that landfills represent an important food source that may partially support some populations [27,30,38,39,40].

Vultures are long-lived vertebrates and are among the most threatened scavenger species, with more than 80% of species listed as Threatened or Near Threatened on the IUCN Red List [41,42,43]. In a transitioning scenario towards a worldwide circular economy model that aims to reduce waste and close landfills, it is imperative to evaluate the effects that these regulations may have on the demographic parameters of the vulture populations that have adapted to feed in these infrastructures [26,38]. Previous studies have demonstrated that a drastic reduction in food availability can increase vulture mobility and decrease the apparent survival of adults [44,45]. However, information is still lacking on whether food reduction in landfills may affect individuals in a population in terms of their age, which could have important demographic consequences given that population dynamics are primarily influenced by adult survival [24].

The aim of the present study was to estimate the age-specific apparent survival and the proportion of transients in a local Griffon vulture (*Gyps fulvus*) population in Central Catalonia (NE Iberian Peninsula), where nearly 3500 individuals were banded over an 11-year period (2012–2022 included) at an open landfill that shifted organic waste management. In mid-2015, in accordance with European policies aimed at closing landfills (Directive 2008/98/EC and Directive (EU) 183 2018/850), a waste treatment center (WTC) was opened to reduce the amount of organic matter that is dumped in this landfill, thereby providing a natural experiment for studying the effects of predictable food shortages on the age-specific apparent survival of griffon vultures. Available data suggest that other environmental factors such as food availability did not change in the area of influence around the landfill since sanitary regulations only allow livestock corpses to be left in the field in ZPAEN areas (protection zones for the feeding of scavenging birds), which are mostly concentrated at high altitude in the north-west pre-Pyrenean and Pyrenean Mountains, and wild ungulates seems to form a low fraction of the diet of the griffon vulture population in this area. A previous study [26] reported a decrease in apparent survival from 82% to 76% between 2012–2018 following the installation of the WTC (which led to an 84% reduction in organic matter from mid-2015 to 2018). This was primarily interpreted as being due to increased permanent emigration as the data was collected from a single site. Apparent survival was estimated for the whole resident local population (i.e., individuals recaptured at least once) without distinguishing between adult and non-adult individuals for whom survival is known to vary (e.g., [46]).

Based on available theory and evidence, we formulated several predictions to be tested in the present study. Capture sessions performed at a single site imply that some captured individuals might opportunistically visit the landfill and do not return (hereafter transients; [47,48]). While transients (i.e., individuals that are never recaptured) are usually accounted for to avoid both lack of fit in the model and underestimation of survival probability, they can also be informative of biological processes occurring at the study site [48,49]. Thus, with declining organic waste, we first predicted that the proportion of adult and non-adult (i.e., juveniles and immatures) transients captured in the landfill would increase, meaning that the site would progressively become less attractive and that newly marked individuals would be more prone to becoming transients after the installation of the WTC [49]. Moreover, PAFS tend to attract and benefit non-adult individuals [15,18,30,50,51,52] since this age fraction is less constrained in their foraging behavior as they are non-breeders and not bound to a specific territory. Consequently, a reduction in food at the landfill may lead to increased density-dependent intraspecific competition at the site, resulting in a higher rate of permanent emigration (reduced site fidelity) since non-adults are less competitive and experienced than adults [52,53,54]. By contrast, adults typically have more restricted home ranges, exhibit stronger fidelity to their breeding territory and surroundings (i.e., are more knowledgeable of their foraging grounds), and are physically stronger and more experienced than non-adults [51,53]. Even though the reduction in organic waste in the landfill may lead to increased density-dependent intraspecific competition, adults are likely to be less affected since they can more easily monopolize resources over young individuals [52,53,54]. Hence, our second prediction is that with decreasing organic matter in the landfill, the apparent survival of adults should be less affected than that of non-adults. Additionally, given that experience and competence are expected to increase with age [6,7], we predict that the apparent survival of immatures should be higher and less affected than that of juveniles after the food shortage. To assess these predictions, we employed a Bayesian hierarchical model with the Cormack-Jolly-Seber (CJS) formulation and age-specific effects to produce estimates of the resident apparent survival probability and the proportion of transients at the studied landfill.

## 2. Materials and Methods

### 2.1. Study Area and Sampling

Capture-mark-recapture (CMR) sessions were conducted in the Orís open landfill (42.07° N, 2.20° E, Central Catalonia, NE Spain), where an organic waste management shift occurred in 2012–2022. The landfill opened in 1995 and currently receives substantial amounts of waste from approximately 70,000 households in the counties of Osona and El Ripollès. Up to 2015, unsorted organic waste and other recyclable materials were dumped directly in the landfill, providing food for a large number of opportunistic species including common ravens (*Corvus corax*), yellow-legged gulls (*Larus michahellis*), and various vulture species, including the griffon vulture [26,40]. In compliance with European directives (Directive 2008/98/EC and Directive (EU) 183 2018/850), a waste treatment center (WTC) was constructed in mid-2015 to reduce the amount of waste dumped in the landfill, in which led to a significant reduction in the organic matter available for scavengers and other species. The amount of organic waste was drastically reduced with the opening of the WTC, decreasing from 17,942 tons in 2012 to 450 tons in 2022 (Appendix A).

Due to its location and the amount of waste it receives, the Orís landfill annually accommodates a significant proportion of the population from the central-northern Catalonia and surrounding areas, including the south of France. It is estimated that the number of vultures visiting the landfill has increased steadily each year (up to 2300 individuals) in line with the growth that this population has experienced in recent decades [26]. A permanent roosting site exists on cliffs above the landfill, and there are breeding colonies from 18 km to over 200 km away from the site.

Vultures were captured using a walk-in trap located approximately 200 m from the landfill. Bait consisting of 30–50 kg sheep and cattle carcasses was regularly supplied to attract the vultures. When captured, vultures were marked with a metal and distance-reading band and age was determined by molt plumage patterns, eye and bill coloration, and the type of feathers on their ruffs [55,56]. To ensure accurate age assignment, photos of vultures taken in the field were examined a posteriori. The CMR sessions were conducted year-round, usually once or twice a month, with an average of approximately 15 sessions per year.

### 2.2. Ageing of Vultures and Sampling Interval

Age was initially assigned according to the calendar year, using five age-classes: 1st calendar year (cy), 2nd cy, 3rd cy, 4th cy, and 5th cy or more [56]. Preliminary analyses (not presented here) using the five age-classes showed that some parameters were non-identifiable, particularly during the first years of the study and for the 2nd–4th cy age-classes. This was due to the low number of vultures identified as 2nd–4th cy ages during the initial years of the study (2012, 2013, and 2014). Therefore, we decided to combine these three age classes into one (see [57]). We pooled these three ages based on the assumption that their biology is similar, given that they are all non-breeders and non-territorial (i.e., low site fidelity). This assumption is supported by the known behavior of wild Griffon vultures in Spain, where they typically recruit into a population during their fifth or sixth calendar year and then become territorial (i.e., high site fidelity, [58]). Individuals of unknown ages, of which the majority were captured during 2012 and 2013 and only comprised 8.5% of the database, were removed from the analyses. For the analyses, we pooled captures into six-month periods (January–June and July–December, two semesters per year). It has been shown that for slow-living species such as vultures, the precision of estimates can significantly improve with larger pooling intervals [59,60,61]. Given the six-month interval, we reassigned ages based on a calendar year that was closer to a biological cycle, in which each individual vulture changes age at the beginning of the second semester. Although in the Mediterranean region mean hatching dates are in spring, nestlings only begin to leave the nest in summer and autumn and so are more likely to be captured in the landfill during the months of the second semester of any given year [62,63,64]. Thus, the age classes used for analyses were as follows: “juveniles”, 1-year-old individuals; “immatures”, 2- to 4-year-old individuals; and “adults”, individuals aged 5 years and older.

### 2.3. Apparent Survival Analysis

We initially tested the goodness-of-fit of the full time-dependent CJS model (i.e., apparent survival and recapture probability vary on each occasion) using U-CARE software v2.2 [65]. As suspected, the data fitted the model poorly (χ^2^ = 413.16, df = 155, *p* < 0.001). Heterogeneity was mainly due to the presence of transients in apparent survival (Test3.SR, *p* < 0.001) but also for trap-dependence in recapture probability (Test2.CT, *p* < 0.001, trap-happiness: z = −6.81), which suggested that our model should account for both effects.

We employed a hierarchical Bayesian state-space Cormack-Jolly-Seber model to estimate age-specific apparent survival and recapture probabilities [66,67,68,69]. The state-space formulation provides a framework for explicitly modeling the biological state and the observation process as realizations of Bernoulli trials. The state process is composed of variables *z_i,t_* and *f_i_*, where *z_i,t_* is a matrix describing the true biological state of individual *i* at time *t*, and *f_i_*, a variable that describes the state of individual *i* on the first capture occasion, being *z_i_*,*f_i_* = 1. The states are modeled as Bernoulli trials in subsequent occasions, considering the product of the survival probability of individual *i* alive at *t* that survives until occasion *t* + 1 and the state at the previous occasion *z_i,t_*. To model transients and residents’ age-specific apparent survival, we applied a time-varying individual age covariate with five categories: four that account for newly-marked (a mixture of transients and residents), and previously marked individuals (i.e., residents) for immatures and adults age classes, and one category for juvenile individuals. Newly-marked juveniles were not differentiated from previously marked ones to avoid unidentifiable estimates of individuals captured for the first time during the first semester of a given year because, in the following semester, juveniles change to immature state and so a ‘juvenile resident’ estimate is then non-estimable for these individuals. Therefore, our residents’ juvenile parameter was confounded with transient individuals. Age covariate values were stored in matrix *A_i,t_*, with *i* = 1, …, *n*, where *n* is the number of individuals and *t* = 1, …, *k* − 1, where *k* is the last capture occasion. Transitions to older ages is deterministic in *A_i,t_* covariate, where ages changes from the second semester. Thus, juveniles can only be in the juvenile state from the second semester and change to immature state during the second semester of the following year, and to the adult state after three years of being immature; in this way we guarantee that no reversion from older to younger ages occurs. The probability of an individual alive at the first occasion *t* that survives until the occasion *t* + 1 is $\phi_{A_{i,t}}$, where *A_i,t_* takes the value of 1 if juvenile, 2 or 3 if a newly-marked or resident immature, respectively, or 4 or 5 if a newly-marked or resident adult, respectively. The state process is formally defined as:(1)$$z_{i,t+1}|z_{i,t}\;\sim \;Bernoulli\;\left(z_{i,t}\times \phi_{A_{i,t}}\right),$$

The information of the observation process is provided by the CMR matrix *y_i,t_*, being *i* = 1, …, *n*, where *n* is the number of individuals, and *t* = 1, …, *k*, where *k* is the last capture occasion, and relates with the state matrix as Bernoulli trials in subsequent occasions with probability of *P_i,t_* (*t* = 2, …, *k*). We utilized an individual covariate to model age-specific immediate trap-response in recapture probability *T_i,t_* (see [68]). The *T_i,t_* matrix contains as many columns as recapture parameters and takes the value of 1, 2 or 3 if the individual *i* at time *t*, was captured at the previous occasion (*t* − 1) as juvenile, immature or adult, respectively, and 4, individuals not captured in the previous occasion. Therefore, we identified a ‘trap-happy’ response if recapture estimates of previously captured juveniles, immatures, or adults were higher than not-previously captured individuals, or ‘trap-shy’ if they were lower. The equation of the observation process is described as follows:(2)$$y_{i,t}|z_{i,t}\;\sim \;Bernoulli\;\left(z_{i,t}\times P_{T_{i,t}}\right),$$

To assess how the shift in waste management at the landfill after the opening of the WTC affected the apparent survival and recapture probabilities, we constrained occasions into three periods: (1) the period before the WTC from 2012 to mid-2015, (2) the first after-WTC period from mid-2015 to 2018, and (3) the second after-WTC period from 2018 to 2022. We chose this three-period structure because it provides more accurate estimates than the time-dependent model for some age classes, as well a clearer trend for before and after the opening of the WTC in relation to the reduction of organic matter available as food for scavengers. In the two after-WTC periods there was a reduction by 84.14% (a period of substantial reduction) and 96.42% (a period of extreme reduction) of organic matter dumped in the landfill and relative to the before-WTC period (Appendix A). Therefore, the apparent survival (*ϕ*) was modeled as follows:(3)$$logit\left(\phi_{A_{i,t}}\right)=\beta_{A,WTC\;}+\varepsilon_{A_{i},{WTC}_{i}},$$
$$\varepsilon_{A_{i},{WTC}_{i}}\;\sim \;Normal\left(0,\sigma_{\phi }^{2}\right),$$
where $\phi_{A_{i,t}}$ is the logit apparent survival probability of the *i*th individual of age *A* captured in *t*th interval, and *β_A,WTC_* is the intercept whose values are the logit mean apparent survival of individuals within each age class *A* during each period of the WTC implementation. $\varepsilon_{A_{i},{WTC}_{i}}$ is an individual random effect that accounts for heterogeneity among individuals of a given age and $\sigma_{\phi }^{2}$ is the variance of logit apparent survival among individuals of each age and period [68,70,71]. When using an individual random effect, we assume that each individual has its own underlying mortality risk (or ‘frailty’). Consequently, individual fitness is considered by incorporating individual variability into the survival estimate. This approach contrasts with age-group-specific classical modelling that assumes that all individuals of a given age are of equal quality [70,72,73]. Thus, the proportion of transients among newly marked individuals for each age class and WTC period was estimated as in [47]:(4)$$\tau_{A_{WTC}}=1-\phi_{A_{WTC\left(x\right)}}^{’}/\phi_{A_{WTC\left(x\right)}},$$
where ${\phi ’}_{A_{WTC(x)}}$ and $\phi_{A_{WTC(x)}}$ are the newly marked and previously marked apparent survival estimates of age *A*, respectively, in a given *WTC*(*x*) period.

Recapture probability (*P*) was modelled as follows:(5)$$logit\left(P_{T_{i,t}}\right)=\alpha_{T,WTC}+\omega_{T_{i},{WTC}_{i}},$$
$$\omega_{T_{i},{WTC}_{i}}\;\sim \;Normal\left(0,\sigma_{P}^{2}\right),$$
where $p_{T_{i,t}}$ is the logit recapture probability of the *i*th individual of age-specific trap-response category *T* on the *t*th occasion, and $\alpha_{T_{i},{WTC}_{t}}$ is the intercept with values indicating the logit mean recapture probability of each previously captured individuals of a given age class and mean recapture probability of individuals not-previously captured. Again, we corrected for heterogeneity among individuals with an individual random effect $\omega_{T_{i},{WTC}_{i}}$ to prevent biased survival estimates [74] where $\sigma_{P}^{2}$ was the variance of logit recapture probability among individuals in each trap-response category and period.

We fitted the model using vague priors, including a uniform distribution (0, 1) for values on the probability scale with logit-1(*β*) and logit-1(*α*), as well as normal distributions (0, *σ*^2^) for the variances of individual random effects with a uniform distribution (0, 10) in standard deviations [68]. Estimates were obtained by sampling from the posterior probability distribution, taking every 10th sample from 95,000 iterations of three chains, following a burn-in period of 40,000 iterations, using Markov chain Monte Carlo (MCMC) algorithm. Analyses were conducted in JAGS [75] implemented through the R package “jagsUI” [76], in R [77]. To ensure convergence, we inspected chains visually by examining posterior density plots and checking the Gelman-Rubin statistic (Ȓ) for each parameter, where values less than 1.1 suggest convergence [78]. All parameter chains exhibited good mixing (Ȓ values < 1.1). Estimates are presented as the mean of the posterior samples and 95% Bayesian credible interval probability (95% BCI).

## 3. Results

A total of 3,414 marked vultures and 1,531 recaptures from 2012 to 2022 were used for modeling. At first capture, 637 were aged as juveniles, 1,104 as immatures, and 1673 as adults (Appendix B). Overall, 66.6% (n = 424) of juveniles, 72.5% (n = 800) of immatures, and 76.2% (n = 1274) of adults were never recaptured, while approximately one-third of juveniles and one quarter of both immatures and adults were recaptured at least once.

Apparent survival of juveniles declined after the opening of the WTC, with a difference of 0.02 (95%BCI: −0.16–0.19) during the first after-WTC period and 0.14 (−0.09–0.38) during the second after-WTC period relative to the before-WTC period (Figure 1). By contrast, immatures’ apparent survival tended to increase by 0.07 (−0.007–0.17) during the first after-WTC period and 0.06 (−0.03–0.16) during the second after-WTC period relative to the before-WTC period (Figure 1). Resident adults’ apparent survival increased by 0.04 (−0.04–0.10) during the first after-WTC period with a subsequent decrease of 0.14 (0.07–0.21) during the second after-WTC period, and 0.17 (0.13–0.21) relative to the first after-WTC period (Figure 1) (Appendix C).

The proportion of transients among newly marked immatures tended to increase by 0.20 (−0.11–0.52) during the first after-WTC period and by 0.13 (−0.15–0.44) during the second after-WTC period, while newly marked adults increased by 0.07 (−0.25–0.42) and 0.24 (−0.12–0.59) during both after-WTC periods (Figure 2) (Appendix C). Recapture probabilities for previously and not-previously captured individuals increased after the WTC was opened (Figure 2). Juveniles’ trap-response was similar to not-previously captured individuals before the WTC opened but changed to ‘trap-shy’ during the first after-WTC period and ‘trap-happy’ in the second after-WTC period. Immatures behaved ‘trap-happy’ before the WTC implementation and ‘trap-shy’ during both after-WTC periods. However, adults were ‘trap-happy’ before the WTC, ‘trap-shy’ during the first after-WTC period, and ‘trap-happy’ in the last after-WTC period (Figure 2) (Appendix C).

## 4. Discussion

We evaluated the age-specific demographic response of a long-lived scavenger species to a drastic reduction in food in a PAFS in terms of its apparent survival and the percentage of individuals that permanently emigrated from the site after the first capture. We used as a natural experiment a local population of ringed griffon vultures that rely on an open landfill where a shift in organic waste management occurred after a waste treatment center (WTC) became operational. This shift caused a progressive reduction from 14,389 to 514 metric tons in the amount of organic matter dumped in the landfill and available as food for vultures during the final four years after the change in waste management (a 96.4% reduction in food availability). In accordance with our first prediction, the proportion of transients among newly marked immature and adult vultures increased over time, thereby indicating that individuals of these ages were more likely to become transients after their first capture due to the diminishing availability of organic matter in the landfill. Regarding our second prediction, the food reduction in the landfill implied a decrease of juveniles’ apparent survival. However, immature residents’ apparent survival increased after the WTC became operational but decreased for adult residents during the final WTC period. Additionally, we explored the immediate trap-response in the three age classes to the reduction in food in the landfill, where each age class behaved differently after the WTC was opened. These findings were made possible by our long-term (11 years) banding effort, which involved approximately 3500 vultures ringed in the landfill. Furthermore, the versatility of the state-space formulation of the Cormack-Jolly-Seber (CJS) model within a Bayesian framework allowed us to fit complex models with individual covariates [68]. This approach enabled us to simultaneously estimate age-specific apparent survival for resident individuals and recapture probabilities while accounting for common sources of heterogeneity in Capture-Mark-Recapture (CMR) studies such as the presence of transients, immediate trap-response, temporary emigration, and individual heterogeneity among age classes [47,79,80].

The proportion of transients among immature and adults increased after the WTC opened and was particularly high for adults, as has been observed in other long-lived bird species [4,49]. Transients are generally high in number when the study area is relatively small as in our case and in long-lived birds such as vultures that exhibit large home ranges [81,82,83,84,85]. As with stressful environmental conditions, food availability also modulates the probability a bird will become transient after the first capture, with a low probability when food is abundant and a higher one when food is scarce. In this way, the local decrease in food can cause an increase of density-dependent effects such as more intense intraspecific competition leading individuals to permanently emigrate after the first capture, this effect being particularly notable in adults [4,49]. Several hypotheses have been proposed to explain the biological meaning of transients in adults, including the cost of first reproduction, a marking and handling effect, or transiting individuals (true transients). The interpretation varies depending on the specific study system and species [48,49]. In our case, the distances from our marking site to nearby griffon vulture colonies range from 18 km to over 200 km. Given that (i) the landfill is a predictable feeding site that attracts a significant number of individuals, including some from distant populations (e.g., some individuals marked in the landfill have come from as far as France and other parts of Spain), and that (ii) vultures are known to visit multiple feeding sites and cover daily foraging distances of 120 to 300 km, a plausible explanation for transients in our system is that it comprises a mixture of true transients (i.e., individuals passing through without belonging to our study population including dispersing non-territorial adults) and individuals from relatively distant colonies prospecting this feeding site.

Our findings suggest that juveniles were negatively affected by the reduction in food at the landfill. Previous studies have demonstrated the positive influence of PAFS on survival rates of several species, with young individuals often being particularly attracted to these sites due to the relative ease of feeding compared to foraging for natural and unpredictable food sources [10,11,12,13,15,18,19]. Typically, after a significant reduction in food availability, it is the juvenile segment of the population that is most affected, resulting in an increased mortality rate or a higher rate of permanent emigration [1,22,23,25]. In our case, however, food reduction was very local and given that juveniles are the most mobile age fraction of the species and not restricted to a specific territory [51,86], the negative apparent survival trend may more likely to be due to a greater permanent emigration rate from the landfill rather than mortality. This suggests a density-dependent effect resulting from reduced food resources, leading to increased intraspecific competition at the site since younger individuals are usually less competitive when compared than their older counterparts; as well, interspecific competition is unlikely as the griffon vulture is typically dominant in the scavenger guild [52,53,54]. Consequently, the landfill may have become less attractive for these young individuals and made them less likely to return.

Contrary to our prediction, the apparent survival of resident immatures did not decrease despite the reduction in available food. Although immatures are generally more experienced and knowledgeable of alternative foraging grounds than juveniles, it is expected that they will also be negatively affected by a food shortage since they are more dependent of PAFS than older vultures, are not strongly bound to a territory, and are usually subordinate to adults when a food source is poor or scarce [15,87]. Other studies of long-lived species exposed to changes in food availability have yielded results that are similar to our findings, which suggests that individual quality within age classes may play a role. For example, [25] observed that after the closure of landfills, the apparent survival of immature Yellow-legged Gulls (*Larus michahellis*) increased in nearby colonies. Similarly, [24] evaluated the long-term dynamic of a griffon vulture colony before-, during- and after-BSE outbreak in density-dependent and density-independent scenarios, and found that immatures’ apparent survival also increased. Both studies and our results suggest that in some systems part of the immature cohort exhibits a demographic response similar to older ages due the disappearance of poor-quality phenotypes of juvenile age. Albeit not restricted to a territory or as experienced as adults, some juveniles may have more knowledge of the foraging grounds and more experience than others, as well a variety of feeding sites within their home ranges that improves their survival probabilities [6,7]. Therefore, among resident immatures, some individuals may behave differently and either revisit the site or permanently leave after the food shortage. Furthermore, the gradual decrease in estimated variances among immature individuals in each period (see Figure A2) suggests that the variability within the resident population lessened after the WTC became operational. This change can be attributed in part to the initial exclusion of transients from residents, as well as the potential persistence of high-quality, experienced individuals that thrive despite the density-dependent effects of a significant food reduction. Hence, a substantial proportion of resident immatures may prefer to frequent and compete for food at the landfill since organic matter may still be enough and available (514 metric tons on average during the last four years of the study) to attract and sustain the large number of vultures that frequent the site each year [26].

The apparent survival of resident adults, on the other hand, partially met our prediction. During the first four years following the food reduction, this parameter remained unaffected and even increased, consistent with findings from prior studies [24,25]. However, during the subsequent four years of extreme food reduction, apparent survival decreased. Although the factors affecting adults are not entirely clear, one reasonable explanation is that the amount of food available at the landfill no longer satisfied the energy requirements of this age class [23]. Thus, the site became less attractive for vultures, especially those nesting at a considerable distance from the landfill that, consequently, shifted their foraging preferences to more viable food sources such as other PAFS and non-predictable (e.g., wild carrion) feeding sites where they can monopolize resources more easily [52,53,54]. Alternatively, this reduction in apparent survival during the second after-WTC period could also be the product of non-natural mortality of individuals. Adults feed more frequently on unpredictable food sources than young birds, which makes them vulnerable to toxic impacts [18,23]. Adult griffon vultures are potentially susceptible to lead ingestion from game animal carcasses [87], veterinary drugs from extensive livestock production, and anticoagulant rodenticides applied in intensive livestock production facilities and landfills [88,89,90,91]. The ingestion of these toxic substances may cause death or induce sub-lethal effects, which may increase the risk of mortality from other causes such as collisions with wind turbines [87,92,93,94]. For example, 28 out of 42 ringed individuals (67%) in the Orís landfill and found dead elsewhere were adults (individuals excluded from this analysis). However, only in nine of them (32%) was their apparent cause of death identified, being primarily due to electrocution, collision with powerlines, and wind turbines (authors’ unpublished data). Recovering and determining the actual cause of death for marked vultures can be challenging due to the difficulty in locating their carcasses, which sometimes may lack markings, thereby rendering them unidentifiable [95,96]. Additionally, when carcasses are found, their partial or extensive deterioration can make necropsy inconclusive, particularly regarding toxic substances. Nevertheless, if non-natural mortality is the cause of the decline of adults’ apparent survival in the landfill, it could have serious consequences for the population dynamics of the species as adult mortality can have a substantial impact on population size [24].

Trap-dependence is one of the most common sources of heterogeneity in recapture probability [97]. The trap-happiness observed in our GOF-test was expected because the captures in this study were made using baited traps [97]. Additionally, the presence of transients in the data (for which *p* = 0, significantly differing from individuals recaptured multiple times), along with the scenario of food shortage [98], further contributed to this expectation. Differentiating recapture probability between previously captured and not-previously captured individuals is a widely used method to account for trap-response derived from capture and handling methods, as well as for temporary emigration such as that caused by the possible seasonal movement patterns of our study population [85]). In our analysis, we employed an individual categorical covariate to explore the age-specific trap-response associated with food reduction in the landfill. This approach allowed us to simultaneously address unmeasured individual heterogeneity by incorporating an individual random effect in recapture probability, thereby producing unbiased survival estimates [74]. Our findings indicate that recapture probabilities increased after the implementation of the WTC for both not-previously captured and previously captured individuals in all three age classes (see Figure 2 and Table A3). This trend suggests that vultures were more likely to be recaptured after the food reduction, possibly because the baited walk-in-trap became more attractive and functioned as a predictable feeding site (extrinsic heterogeneity due to the capture method). This shift in behavior has been observed in small mammals [98]. Furthermore, the immediate trap-response for each age class varied during the three periods evaluated and suggests that the trap-response initially diagnosed with the GOF-test (i.e., trap-happiness), may not fully reflect the full nature of this phenomenon. Instead, trap-response can be dynamic in a system, with individuals within groups (e.g., age classes) displaying shifts from ‘trap-happy’ to ‘trap-shy’ and vice versa over time [98]. For instance, some immature and adult individuals were ‘trap-happy’ before the WTC was opened. However, during the next two after-WTC periods, all three ages behaved differently and were ‘trap-shy’ during the first-WTC period, and ‘trap-happy’ for juveniles and adults, and ‘trap-shy’ for immatures during the second-WTC period. Unraveling the reasons for this variability are beyond the scope of our study due to the lack of individual covariates other than age, which could explain part of this heterogeneity during modeling (intrinsic heterogeneity, such as sex, mass, or personality) [73,98], and the absence of temporal environmental covariates that may modulate trap-response at the site (e.g., landfill machinery activities and daily food regimes). However, it is important to emphasize that trap-response can be highly variable over time and among individuals within each age class.

## 5. Conclusions

In summary, our study provides valuable evidence of the age-specific detrimental impacts of food reduction on the demographic parameters of a long-lived scavenger bird species specialized in feeding on predictable anthropogenic food subsidies. From a conservation perspective, the closure of landfills is a desirable objective that will reduce the adverse effects on both the environment and the species that rely on this food resource. These effects are evident at the level of life history traits [17], health [34,35,87], and human-wildlife conflicts [99]. While our study suggests that the observed negative effect may be largely associated with permanent emigration due to a diminished food supply, it is worth noting that European regulations calling for the reduction of food in landfills are being implemented across the whole of Catalonia (Appendix D, Figure A4). This synchronized reduction in food resources could have significant consequences for demographic rates and ultimately threaten an important food source for numerous species that rely on these facilities and, in particular, for the griffon vulture population in the northeast Iberian Peninsula. These vultures are highly specialized and feed on carcasses originating from extensive livestock farming and landfills [26,38] and younger age classes being particularly dependent on landfills, especially during periods of wild food scarcity such as winter [27,28,30,31,100]. As a result, the reduction in food and the potential closure of landfills may lead to a dramatic shift in trophic strategy among scavenger species, forcing them to rely more on less predictable food sources (and, to some extent, predictable ones like supplementary feeding sites for scavengers), as previously predicted for Egyptian vultures (*Neophron percnopterus*) [40]. To mitigate these impacts, conservation measures should be considered, such as the establishment of a network of randomly distributed supplementary feeding stations to encourage natural foraging [101] or the expansion of protection zones for the feeding of scavenging birds (known as ZPAEN zones) in areas that are currently underrepresented [102]. These efforts should be pursued at a regional level to prevent undesirable negative effects on the population dynamics of these species.

## Figures and Tables

**Figure 1 animals-13-03529-f001:**
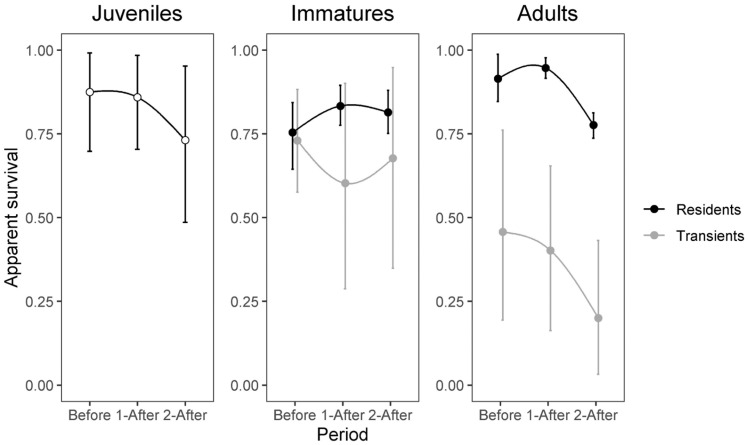
Apparent survival of juveniles, immatures, and adults before (2012–mid 2015), during (mid 2015–2018) and after (2018–2022) the waste treatment center (WTC) was opened. Estimates for newly marked (“Transients”) and previously marked (“Residents”) individuals are only differentiated for immatures and adults (see Methods). Error bars represent the 95% Bayesian credible interval.

**Figure 2 animals-13-03529-f002:**
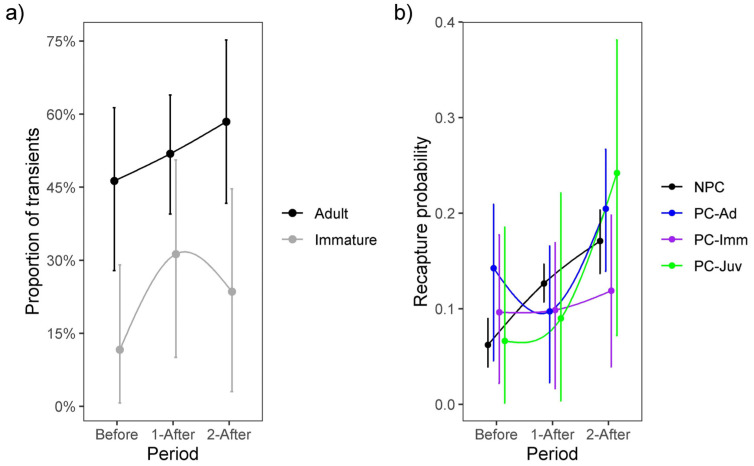
(**a**) Proportion of transients among newly-marked immatures and adults, and (**b**) recapture probability accounting for age-specific trap-response and individual heterogeneity, before (2012–mid 2015) and during the two periods after (mid 2015–2018 and 2018–2022) the waste treatment center (WTC) was opened. NPC = not-previously captured individuals, PC-Ad = previously captured adult individuals, PC-Imm = previously captured immature individuals, and PC-Juv = previously captured juvenile individuals. Error bars represent the 95% Bayesian credible interval.

## Data Availability

The data presented in this study are available in this article and from the corresponding author upon reasonable request.

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
