# Peer review of "Age-Specific Demographic Response of a Long-Lived Scavenger Species to Reduction of Organic Matter in a Landfill"

_animals, 2023, doi:10.3390/ani13223529_

Round 1

Reviewer 1 Report

Comments and Suggestions for Authors

The manuscript explores the effects of food reduction on the age-specific demographics of Griffon vultures, utilizing long-term data. The article is well-structured, well-written, and offers novel insights into understanding how the vulture population changed after the implementation of the waste treatment center.

I do have some minor concerns to address:

1.     How many vultures were marked and recaptured before and after the waste treatment center was implemented?

2.     What about the migration pattern of Griffon Vultures? Since this species is migratory, it's important to consider how the movement of juveniles, immatures, and adults may have affected the sightings of juveniles at the landfill site.

3.     Is the open landfill the only landfill site in the area, or are there others? If there are other landfill sites, how far away are they? Vultures visiting the open landfill might not have visited other landfills, which could impact the capture and recapture process.

4.     Did you also observe the presence or density of other scavengers after the implementation of the Waste Treatment Center? Are their trends similar? These scavengers could potentially increase competition with juveniles, ultimately affecting the juvenile population.

5.     The distance between the roosting sites or colonies and the landfill sites should be mentioned in the study area section.

Reviewer 2 Report

Comments and Suggestions for Authors

In this study, the authors investigated changes in demographic parameters for a population of griffon vultures following the partial closure of a local landfill. This is an important topic as it can inform wildlife managers about potential consequences of landfill closure for threatened or vulnerable species as it appears to be a recent management policy in many countries. I am not particularly familiar with CMR modelling, but it appears sensible. I like the use of random effects, the partition of the population in three age classes and the use of three time periods. The paper is clearly written and the relevant literature is used. I have three general comments and some minor ones listed below.

General comments

How representative is the population of vultures using the bait site? I am wondering whether there are segments of the nearby population pool that never use or visit the bait site. If this is the case, the mortality pattern might not be representative of the whole population. Is this known for this vulture population or for other populations of this species or other species? Similarly, is it known whether some individuals never visit baited walk-in traps?

I agree that this is a natural experiment with a before and after scenario, but is it possible that other variables changed during this period? I am thinking in particular of the weather or the overall food availability at the regional level. This might have changed substantially over the duration of the study period and could account for some of the results. Also, some age classes might be more susceptible to these changes. Is this likely?

I was wondering whether it is possible to compare juvenile, immature and adult survival with that of other non-provisioned populations of this species. This seems relevant because I was surprised to see that apparent survival was quite low after the partial closure of the landfill. Surely, for such a large species, apparent survival should be closer to 90% or above, no? This may tell us something about the value of doing CMR near baited sites.

Minor comments

Line 196: Can you expand a little on full-time dependence? Not all readers might be familiar with this type of model.

Line 393 : More accurately, given the results, their survival did not decrease.

Comments on the Quality of English Language

Just minor slip-ups here and there.

Reviewer 3 Report

Comments and Suggestions for Authors

Brief summary

This paper covers variation in age-cohort demographics of an avian scavenger (vulture) population  faced with a dramatic (>95%) reduction in food availability caused by changed management practices at a landfill in northeast Spain.

General concept comment

Overall, a well written manuscript, with sound analyses, and some interesting, occasionally counter-intuitive results. Some of these may be due to the study setup whereby food is provided to trap animals (to derive age-specific demographic data) in a study focused on assessing the impact of reduced food availability. This ‘conundrum’ is acknowledged and satisfactorily dealt with by the authors.

Comments on the Quality of English Language

Some of the English in the discussion is ‘awkward’ and I have made some suggestions that will improve the flow and readability.

Line 82 – replace ‘fractions’ with cohorts

Line 83 – try and explain why natural food supply is reduced in winter – often mortality rates of prey species (potential carrion e.g. ungulates) would be higher in winter

Line 271 – replace ‘me’ with we

Line 292 – replace ’A’ with Overall, 66.6%

same sentence, perhaps explain these figures as 1/3  of juveniles and ¼ of both immatures and adults ‘acquired’ resident status

line 338 – Maybe state/emphasise that this is a 96.4% reduction in food availability

line 348 – replace ‘implantation’ with ‘became operational’

line 365/6 – replace ‘being’ with ‘with’

line 368 – insert ‘a marking and handling’ effect

line 387 - replace 'unbounded’ with ‘not restricted’

line 395 - replace 'the’ with ‘alternative’

line 396 – insert 'they’ [since they are]

line 397 – replace strong with strongly

line 399 – replace ‘shifts’ with ‘changes’

line 401 – replace ‘near’ with ‘nearby’

line 406 – insert ‘the immature cohort’

line 407 – rephrase end of line to ‘they are not restricted’

line 418 – replace ‘part’ with ‘proportion’

lines 428/9 – suggest rephrase ‘especially those nesting at a considerable distance from’ the landfill

line 434 – replace ‘ages’ with ‘birds’ and ‘intoxication’ with ‘toxic impacts’

line 437 - replace 'toxics’ with ‘toxic substances’

line 447 – see above re intoxications

line 456/7 – suggest ‘captured and uncaptured’ individuals

line 467 – replace ‘was’ with ‘has been’

line 492 – it is (not it’s)

line 497/8 – delete (second) ‘population’
